# Pregnancy Outcomes in SARS-CoV-2-Positive Patients: A 20-Month Retrospective Analysis of Delivery Cases

**DOI:** 10.3390/medicina59020341

**Published:** 2023-02-10

**Authors:** Andreea Moza, Elena S. Bernad, Diana Lungeanu, Marius Craina, Brenda C. Bernad, Lavinia Hogea, Corina Paul, Cezara Muresan, Razvan Nitu, Daniela Iacob

**Affiliations:** 1Department of Obstetrics and Gynecology, “Victor Babes” University of Medicine and Pharmacy, 300041 Timisoara, Romania; 2Clinic of Obstetrics and Gynecology, “Pius Brinzeu” County Clinical Emergency Hospital, 300723 Timisoara, Romania; 3Center for Laparoscopy, Laparoscopic Surgery and In Vitro Fertilization, “Victor Babes” University of Medicine and Pharmacy, 300041 Timisoara, Romania; 4Center for Modeling Biological Systems and Data Analysis, “Victor Babes” University of Medicine and Pharmacy, 300041 Timisoara, Romania; 5Department of Functional Sciences, Faculty of Medicine, “Victor Babes” University of Medicine and Pharmacy, 300041 Timisoara, Romania; 6Department of Neuroscience, “Victor Babes” University of Medicine and Pharmacy, 300041 Timisoara, Romania; 7Center for Neuropsychology and Behavioral Medicine, “Victor Babes” University of Medicine and Pharmacy, 300041 Timisoara, Romania; 8Department of Pediatrics, “Victor Babes” University of Medicine and Pharmacy, 300041 Timisoara, Romania; 9Clinic of Neonatology, “Pius Brinzeu” County Clinical Emergency Hospital, 300723 Timisoara, Romania

**Keywords:** SARS-CoV-2, COVID-19, vaccination, pregnancy, delivery, adverse outcome, obstetrical outcome, neonatal outcome

## Abstract

*Background and Objectives*: The SARS-CoV-2 infection brings supplemental risks for pregnant women. Due to controversial hesitancy, their vaccination rate was lower in 2021 compared to the general population. In addition, access to maternal care was reduced during the pandemic. We conducted a retrospective cross-sectional analysis of the health records data over 20 months (1 April 2020 to 20 November 2021) aiming to explore the outcomes in SARS-CoV-2-positive cases referred for delivery to a tertiary public hospital in Western Romania. *Materials and Methods*: Women with SARS-CoV-2 infection diagnosed for the first time at the moment of birth who delivered singletons after 24 weeks of gestation, and had a clear immunization status were included in the analysis. *Results*: Out of the 97 patients included in the study, 35 (36%) had undergone ARN-based vaccination. Five cases of maternal death were recorded (all unvaccinated). Our retrospective exploratory analysis showed that the presence of COVID-19 symptoms in the SARS-CoV-2-positive patients made a significant impact on the delivery hospitalization, with a median hospital stay increase from 5 to 9 days (Mann–Whitney test, *p* = 0.014): longer hospitalization was recorded in the symptomatic cases irrespective of their vaccination status. No other adverse outcomes, such as gestational age at delivery, C-section rate, 5 min Apgar index, or birth weight were associated with the presence of symptoms. *Conclusions*: Our clinic maintained safe maternal care for the COVID-19 patients during the analyzed period. Vaccination of the expectant women was beneficial in SARS-CoV-2-positive patients by lowering the risk of COVID-19 symptoms, with subsequent implications on the newborns’ health and maternal attachment.

## 1. Introduction

COVID-19, a coronavirus disease that has spread globally, increasing morbidity and mortality, has not eluded women of reproductive age. Studies suggest a diminution of fertility in infected women, although it seems to be reversible so far, making pregnancy a more susceptible condition in cases of COVID-19 illness in the women of reproductive age [1,2,3]. There was an increased contextual risk of infection for pregnant women illustrated by the governments’ response stringency index (GRSI) or the overall health system performance as measured by the World Health Organization [4,5].

COVID-19 can seriously affect pregnant women, who are acknowledged to be at higher risk for severe disease when they are symptomatic [1,2,4,5,6]. Pregnancy complications can occur in COVID-19 patients, such as maternal or fetal death, pre-term birth, or caesarian section (C–section) due to the maternal medical condition. According to available evidence, pregnant women are especially vulnerable to severe COVID-19 complications like acute respiratory distress syndrome, acute renal failure, thrombo-embolic events, and other unfavorable cardiac events [7]. As a result, these women are more prone to invasive ventilation, hospitalization in an intensive care unit (ICU), or to extra-corporeal membrane oxygenation [8]. Infected pregnant patients in poor health may require induction of labor, resulting in premature birth with all of its consequences [9].

While large systematic reviews fail to demonstrate vertical transmission, some papers (mostly case reports) shed some light on this subject [10,11,12].

Most congenital infections are asymptomatic, but cases of neonatal pneumonia, thrombocytopenia, altered liver function, feeding difficulties, cardiac arrhythmia, and thrombosis were reported. Furthermore, retinal and choroidal abnormalities in the newborn have also been reported [12,13,14,15]. In spite of this, even in the absence of vertical transmission, maternal infection can have an indirect effect on the developing fetus. Histopathological examinations of the placenta have shown vascular abnormalities, particularly in the decidua, as well as thrombi in the fetal arteries, alterations that can determine fetal hypoxia [16].

Vaccines have been developed, and national as well as international authorities, organizations, or health agencies all over the world have recommended the RNA-based vaccines for pregnant or lactating women: Comirnaty (developed by BioNTech, Germany and Pfizer, USA), and Spikevax (previously known under the name of the manufacturing company, Moderna) [17,18,19,20]. Despite the fact that vaccination did not offer full protection against SARS-CoV-2 and the infection and the occurrence of high rates of hesitation (particularly due to the uncertain influence on the fetus and the woman’s future fertility), promising results have been reported regarding vaccine effectiveness [21,22,23]. In addition, prevention of hospitalization was reported in vaccinated pregnant patients, with no additional pregnancy-related risks [22,24,25,26,27,28]. All over the world, 120 countries have recommended COVID-19 vaccine in pregnancy. Along with other 62 countries, Romania’s position was and remained that COVID-19 vaccination of pregnant or lactating women is ”permitted” as of 10 November 2022 [29].

During the first three waves of the COVID-19 pandemic, the Department of Obstetrics and Gynecology of the Clinical County Hospital of Timisoara, a tertiary hospital affiliated with the “Victor Babes” University of Medicine and Pharmacy, hospitalized all COVID-19 patients from Western Romania with obstetrical or gynecological problems [30]. While effectiveness of the health services is a complex challenge encompassing multicriterial aspects, finding the workable practices appropriate to the pandemic crisis increased the demands on the hospital’s management and every healthcare worker, especially the young trainees [31,32]. In this worldwide pandemic crisis, our hospital as a whole, and staff at all levels, faced disruptive challenges to their efforts to continue providing the healthcare services and preserving safe maternity and neonatal care and paying attention to all pregnant women, especially to those with chronic pathologies or pregnancy-related comorbidities [33,34,35].

Taking all the above into consideration, we conducted a retrospective exploratory analysis on the patients‘ health records aimed at debriefing the clinical activity’s effectiveness in providing seamless obstetrical services for SARS-CoV-2-positive patients. This secondary data analysis covered both the initial period, when no vaccine was available, and the first 10 months after vaccines became available to the general population in Romania. In this latter period, professional organizations recommended vaccination for pregnant women. General maternal and neonatal outcomes were used as performance indicators in this study. Maternal outcome was measured in the study using the following parameters: severity of lung disease on admission on computer tomography, need for intensive care, intubation and extracorporeal membrane oxygenation, and maternal death. The rate of c-section due to severe maternal disease and the rate of prematurity, as well as the rate of intrauterine demise, were used to evaluate the neonatal outcome. The possibility of vertical transmission of the coronavirus was also investigated.

## 2. Materials and Methods

### 2.1. Study Design and Population

This was a retrospective study with cross-sectional design encompassing the first three waves of the pandemic: 1 April 2020–20 November 2021. It occurred in a single tertiary medical center, the Department of Obstetrics and Gynecology of the Clinical County Hospital of Timisoara, Romania, which was one of the facilities that by the decree of the Health Ministry of Romania hospitalized all infected COVID-women with obstetrical or gynecological problems from west side of the country.

During the time frame of the study, there was a period (beginning on 1 April 2020 and ending on 31 May 2020) in which the Obstetrics and Gynecology Department of the Clinical County Hospital of Timisoara admitted only patients with confirmed SARS-CoV-2. After this period of time, the admittance of uninfected patients was allowed, but in a restricted number, and the unit was primarily focused on the management of SARS-CoV-2-infected patients. In this timeframe, 397 women were discharged after receiving treatment for SARS-CoV-2 infection. Some of these patients had a gynecological health issue, some were discharged while pregnant, and others were discharged after delivery. 278 patients were discharged from the hospital in the postpartum period, out of a total of 397 patients.

The purpose of the study was to investigate the outcomes of women with SARS-COV-2 infection at the time of birth as well as whether immunization status affected the outcomes of the mother and the newborn. Data were retrieved from patients’ records.

The decision to query the hospital’s data base utilizing the discharge diagnosis as the primary criteria was made in order to include the infected patients who required urgent delivery in other institutions before laboratory confirmation. These patients, as well as their newborns, were transferred to this referral department as soon as the SARS-CoV-2 RT-PCR test was available (transfer usually took place less than 8 h after delivery).

Only women who had singleton pregnancies and delivered after 24 weeks of gestation were included in this study.

Patients whose records do not mention their immunization status or who had prior SARS-CoV-2 infection were excluded from the study. After applying the exclusion criteria there were a total of 97 individuals who were eligible for the study. All patients were Caucasian.

The flow diagram of this study is depicted in Figure 1.

### 2.2. Outcome Measures

The analysis included the patients’ socio-demographic data and general characteristics, such as age, weight, weight gain during pregnancy, smoking status, number of previous pregnancies and deliveries, and the duration of hospitalization. Maternal comorbidities and management, or COVID-19-related complications (severity of lung disease, admission in the intensive care unit, ECMO, intubation requirement, and maternal death) were also included. Recorded information on paraclinical investigations was included when available (missing data was an issue for this information). No imputation was done for the missing data.

Obstetrical and neonatal outcome metrics included the following indicators: gestational age at delivery, gender of the newborn, maternal death, intrauterine fetal demise (IUFD), C-section (overall and due to COVID-19 complications), birth weight, 5 min Apgar index for the newborn, and whether or not the newborn had COVID-19.

### 2.3. Data Analysis

Descriptive statistics included the observed frequency counts (percentage) for categorical variables and median (interquartile range with Tukey’s hinges) for numerical variables. For the maternal age, the mean and standard deviation were also included in the descriptive table. All numerical variables were non-normally distributed (Shapiro–Wilk statistical test was employed for checking the normality of the distributions). Univariate non-parametric statistical tests were applied to compare the distribution of numerical data across two or multiple groups, as appropriate (either Mann–Whitney U or Kruskal–Wallis tests, respectively). The chi-square statistical test (either asymptotic, Fisher’s exact test, or Monte Carlo simulation with 10,000 samples) was applied to check the statistical significance of the association between the categorical variables.

The statistical analysis was conducted at a 95% level of confidence and a 5% level of statistical significance. All reported probability values are two-tailed.

Statistical analysis was performed with the statistical software IBM SPSS v.20 and open-source R v.4.0.5 packages.

## 3. Results

### 3.1. COVID-19 Symptoms in SARS-CoV-2-positive Women

Sixty-two patients (64%) were unvaccinated, and all of them had COVID-19 symptoms at hospital admission. Respiratory symptoms were the most commonly reported by unvaccinated patients (40/62). These included rhinorrhea, sore throat, cough, and dyspnea. Eleven patients showed neurologic involvement of SARS-COV-2 infection, including ageusia, anosmia, and agnosia. Twenty-one patients reported malaise. Ten patients had fever, and three had vomiting. Out of the 35 women who underwent at least one dose-vaccination, 12 (34.29%) had COVID-19 symptoms. Most frequent symptoms in vaccinated women were rhinorrhea (10/12), followed by neurologic symptoms (5/12), cough (3/12), and dyspnea (1/12).

All vaccinated patients had received Comirnaty (developed by BioNTech and Pfizer).

Overall, the difference in proportion of symptoms was highly significant: Monte-Carlo-simulated chi-square test, *p* < 0.0001. Figure 2 illustrates the distribution of the patients in the two groups corresponding to the vaccination status, with their respective number of symptomatic and asymptomatic cases.

### 3.2. Maternal Characteristics

Table 1 presents the socio-demographic and clinical characteristics of the 97 patients included in the analysis. Missing data were specified when this issue occurred.

The duration of hospitalization varied among the patients, ranging from 3 to 33 days in unvaccinated patients, between 3 and 22 days among symptomatic vaccinated patients, and no more than 15 days in asymptomatic vaccinated patients. Overall, asymptomatic patients had a shorter hospitalization compared to the symptomatic ones, irrespective of their vaccination status (Mann–Whitney statistical test; *p* = 0.032).

There were no differences concerning the other characteristics of the patients, except for the initial weight and their body mass index: the vaccinated pregnant women in the study were significantly heavier.

Table 2 shows the comorbidities and treatments of the study patients, including the vitamin and iron supplements. Although we found no differences between the three groups, a high percentage of patients with at least one comorbidity is noticeable.

### 3.3. Laboratory Findings in SARS-CoV-2-Infected Patients

Table 3 presents the results of paraclinical investigations at admission.

### 3.4. Management of COVID-19 Disease

On computer tomography, 14 (22.58%) unvaccinated patients had signs of lung lesions at the time of admission. The percentage of affected lung parenchyma ranged from 15% to 80%: 15% (1 patient), 20%(2 patients), 30%(2 patients), 35%(1 patient), 40%(4 patients), 60%(2 patients), 70%(1 patient), and 80%(1 patient). In the immunized group, one patient displayed signs of pulmonary deterioration, with 30% of lung parenchyma being affected by COVID-19.

Table 4 presents the summary statistics of COVID-19 complications.

Low levels of oxygen in the blood were treated with high flow oxygen therapy or extracorporeal membrane oxygenation if necessary. Due to their poor health condition, 13 women were admitted to the intensive care unit (ICU). One of them was immunized against COVID-19, but all the other were not. Seven (11.3%) of the unvaccinated women were placed on high flow oxygen therapy, and two (3.2%) were placed on extracorporeal life support. In seven cases, the deterioration of the patient’s health entailed intubation (all belonged to the unvaccinated group). Endo-tracheal intubation lasted from 2 to 29 days. In contrast, none of the women in the vaccinated group needed intubation.

### 3.5. Pregnancy Outcomes

Gestational age at the moment of delivery varied from 25 weeks to 42 weeks. The overall percentage of prematurity was 25.7% (25 out of the 97 subjects in the study). Table 5 shows the obstetrical and neonatal outcomes in these 97 subjects.

More than half of the patients underwent a C-section delivery, and for many of them the surgical solution was chosen based on their medical condition, namely due to the COVID-19 complications.

Severe obstetrical outcomes occurred in the unvaccinated patients: there were 5 cases of postpartum maternal death due to COVID-19. All had anemia, but 2 out of the 5 patients also had additional comorbidities: thyroid disease and pregnancy-induced hypertension (1 patient) and gestational diabetes (1 patient). These patients’ ages ranged from 19 to 37 years, with 3 of them being over 35. In most of them (4/5), the infection occurred during the third trimester (33–38 weeks of gestation), but one of the patients had a 25-week pregnancy. All of them were admitted to ICU and received high flow oxygen therapy. In all cases, delivery was recommended due to the mother’s severe COVID-19 disease, resulting in live neonates. All neonates tested negative for SARS-CoV-2 infection at birth and remained that way for the duration of their hospitalization.

Among the studied 97 patients, there were 3 cases of intrauterine fetal death/demise (IUFD) in the third trimester, at 28, 30, and 31 weeks of gestation. All stillbirths were diagnosed at hospital admission and occurred in unvaccinated patients.

No evident risk factors for IUFD were detected in any of the cases, but one pregnant woman arrived to the hospital with severe COVID-19 disease (computer tomography showed 70% lung destruction) . Pathology reports were not available.

There were two cases of COVID-19 infected newborns, both of unvaccinated mothers. They were born at 37 and 38 weeks of gestation, respectively. Due to the risk of uterine rupture on a previous uterine scar, one was delivered via C-section, whereas the other was delivered vaginally. The neonates were asymptomatic at birth and progressed well during their hospitalization.

## 4. Discussions

This analysis described the pregnancy outcomes of 97 patients who had a positive RT-PCR test for SARS-CoV-2 infection at the time of delivery. The main objective of this study was an exploratory assessment of maternity care over a period of 20 months of the COVID-19 pandemic. It also aimed to ascertain the supposed difference between the vaccinated and unvaccinated SARS-CoV-2-positive patients in regard to the pregnancy outcomes.

The rate of vaccination in our sample of pregnant women was lower than in the general population. This might be explained by Romania’s position on COVID-19 vaccination while pregnant or lactating, which was less stringent than that of other countries [4,29]. Because of the small size of the study lot, we were unable to determine the prevalence of symptoms among the vaccinated; however, a recent study conducted in New York City reported that 21% of patients were symptomatic [36].

Due to the specific alteration of the immune system, pregnant women are considered to be less likely to develop symptoms if infected, but factors such as BMI > 25, presence of chronic health conditions, advanced maternal age, or belonging to socially disadvantaged groups are known to be associated with symptomatic COVID-19 disease [37,38,39,40]. In the analyzed period, there were five cases of maternal death, all in the unvaccinated patients. Three of these women were of advanced maternal age.

We found that the ICU rate was also associated with the unvaccinated status, but the numbers were too low to reach statistical significance. Only one vaccinated patient needed to be admitted to an intensive care unit due to a severe infection; all the other 11 patients had mild symptoms. On the other hand, the study design and the probable lack of statistical power prevented us from having a straight answer to possible consistency with other findings that reported no relation between their vaccination status and ICU admission [41]. Lowering the risk of complications entails consistent improvement of the chance for mother–child attachment [42].

The risk of infection was reported to decline quickly after the first vaccination dose (in 10 days), and maternal antibody response should be taken into account after vaccination [43]. Prabhu et al. reported specific immunoglobulin IgM and IgG in 71% of cases, only IgG in 16% of cases, and no immunoglobulin was found in 13% of patients [44]. Our secondary data analysis did not allow retrieving comprehensive information on the vaccination scheme (i.e., whether completed or not). Worldwide fatality rates among pregnant patients infected with SARS-COV-2 varies widely [1].

While some studies reported a very low (0.2%) mortality rate in pregnant patients, other studies reported a death rate as high as 12.7% (1858/15105 pregnant patients) [45,46]. These wide margins might be related to the national stringency regarding the pandemic measures and to the overall health systems’ performance and level of national income [1,4,5]. In Romania, COVID-19 vaccination while pregnant or lactating was “permitted”, as opposed to highly recommended in other countries; therefore, one would expect a lower rate of vaccination in pregnant women compared to the general population [30].

The current literature on vertical transmission of the COVID-19 infection is still in the early stages. Despite evidence of placental histology and newborn immunological blood tests, the rate of reported congenital COVID-19 infection is very low, ranging from 3.2% to 8.40% [11,47,48]. In a previous study, our team also reported cases of possible vertical transmission [49]. When the newborns experience COVID-19 symptoms, they are likely to require respiratory support or to be transferred to a neonatal ICU, particularly if their mothers are symptomatic, too [50].

Despite the low rate of vertical transmission, poor neonatal outcomes such as intrauterine fetal growth restriction or fetal distress was more frequently found in case of maternal COVID-19 disease [51,52,53]. While poor maternal oxygenation could be blamed for this issue, most studies point to the abnormal functionality of the fetoplacental unit [54]. Abdel Massih’s review on SARS-COV-2-associated placental insufficiency concluded that 20% of the infected neonates had signs of intrauterine hypoxia. Moreover, placental damage was observed negative RT-PCR newborns, probably as a response to maternal cytokine storm [55]. When compared to uninfected cases, histologic findings suggestive of vascular malperfusion such as chorioangiosis, intramural fibrin deposition, vascular ectasia, and perivillous fibrin deposition were strongly associated with SARS-CoV-2-positive pregnancies [54]. Additionally, thrombosis of the fetal chorionic plate and decidual arteries were more commonly seen in the placentas of infected women [54].

Studies show vaccination in pregnancy benefits both the fetus and the newborn, as a result of the passive transplacental passage of antibodies. Upon sampling blood from the umbilical cord of vaccinated women after delivery, Collier et al. identified IgG antibodies against the receptor binding domain (RBD) of SARS-CoV-2, against S protein as well as neutralizing antibodies [40]. Nonetheless, the detection of antibodies in the newborn’s blood depends on the number of the vaccine doses (43.6% vs. 98.5%), as well as on the latency between the last dose and the birth (the longer the latency, the higher the transfer) [40,56]. In the current analysis, 2 out of the 59 live newborns of unvaccinated mothers tested positive for SARS-CoV-2 infection. Among the vaccinated patients (35 in total), none tested positive. We could not confirm the congenital viral infection according to the WHO recommendations of RT-PCR on the placental tissue or detections of specific viral particle in the placenta (possible only with an electronic microscope); on the other hand, the neonates were sampled right after birth and the attending staff took strict precautions to avoid contamination [57].

### Limitations

The main limitation of this analysis stems from the cross-sectional design and the secondary use of data from the patients’ records, with limited information and multiple missing values. Conceptualized as an exploratory analysis, no a priori statistical power analysis was conducted; therefore, the risk of type II statistical error (i.e., the false negative rate) in some findings remains a caveat.

Additionally, the analysis comprised limited data from one single clinic, but this is a tertiary public maternity affiliated to the largest university of medicine and pharmacy from the western part of Romania, to which most complicated or problematic cases in the region are referred.

Nevertheless, despite its limitations, this timely report contributes to the growing body of evidence in regard to the benefits of vaccination for pregnancy outcomes.

## 5. Conclusions

Our clinic provided safe maternal care for pregnant COVID-19 patients during the analyzed pandemic period.

The results reported in the present study suggest that vaccination of the expectant women has a beneficial role in SARS-CoV-2-positive patients by lowering the risk of COVID-19 symptoms, with subsequent favorable implications on the newborns’ health and maternal attachment.

Apart from the undeniable limitations, the strength of this debriefing analysis consists of its contribution to the medical evidence towards the benefits of vaccination in pregnancy. It also confirmed the robustness and effectiveness of the maternity services in this region during the first pandemic waves, when the healthcare systems were heavily disrupted worldwide.

The hard lessons learnt during the COVID-19 pandemic can help to improve maternity care, and their promotion may mediate still pending changes in health policies. They can inform decision making at all levels in regard to prioritizing limited resources in the context of the national stringency of public health policies and also guide individual pregnant women in their personal decision about COVID-19 vaccination or vaccination in general.

## Figures and Tables

**Figure 1 medicina-59-00341-f001:**
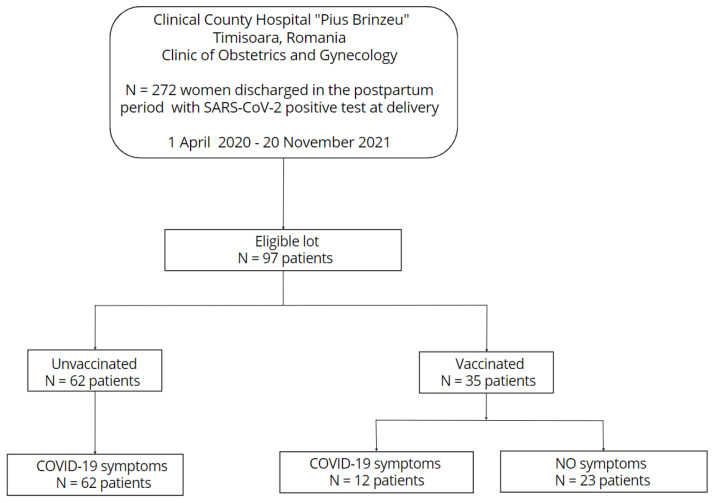
Flowchart of the cross-sectional data collection in this exploratory analysis. 272 women with a positive PCR RT test for COVID-19 were discharged in their postpartum period from the Department of Obstetrics Clinical County Hospital of Timisoara throughout the research period (1 April 2020–20 November 2021). In every instance, testing for the SARS-CoV-2 infection took place prior to delivery. There were 97 suitable patients left after the exclusion criteria were applied (the presence of multiple pregnancies, gestational age at birth less than 24 weeks, and unclear immunization status against SARS-CoV-2 infection). Out of these, 35 patients were immunized against COVID-19 disease, while 62 patients were unvaccinated (all of them developed symptoms), and 12 of the vaccinated individuals showed signs of infection at the time of birth, whereas the remaining patients were asymptomatic.

**Figure 2 medicina-59-00341-f002:**
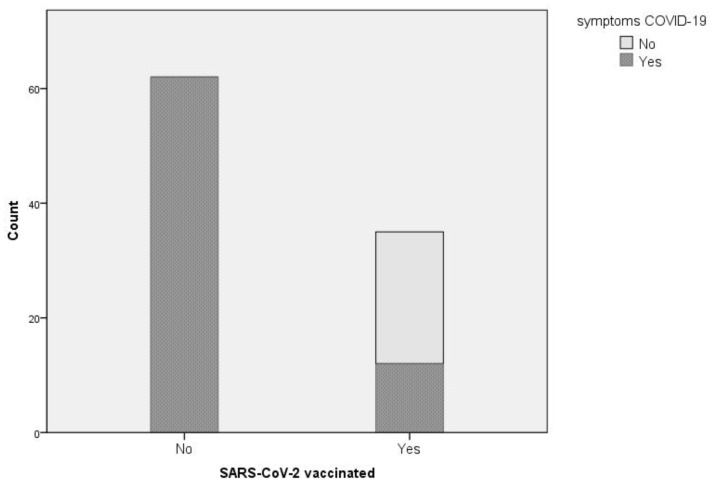
COVID-19 symptoms observed among the SARS-CoV-2-positive patients.

**Table 1 medicina-59-00341-t001:** Socio-demographic and clinical characteristics of the pregnant patients at admission, stratified by vaccination and symptomatic status.

Patient Characteristics	SARS-CoV-2-Positive Patients			
	Unvaccinated	Vaccinated		
	Symptoms +	Symptoms +	Symptoms −	*p*-Value ^(b),(c)^
Variable ^(a),(b),(c)^	N = 62	N = 12	N = 23	
Maternal age [years] ^(a),(b)^	29.90 ± 5.51	27.83 ± 3.38	27.61 ± 4.28	0.077
	30 (26–35)	26.5 (25–30)	27 (25–30)	
Hospitalization days ^(b)^	7 (5–13)	9 (6–10.5)	5 (4–7)	0.032 *
	^(d)^ COVID-19 symptoms vs. non-symptoms, *p* = 0.014 *			
Rural area ^(c)^	24 (38.7%)	6 (50%)	7 (30.4%)	0.531
Gesta ^(b)^	2 (1–2)	1 (1–2)	1 (1–3)	0.376
Para ^(b)^	2 (1–2)	1 (1–2)	1 (1–2)	0.458
Smoker ^(c)^	11 (17.7%)	5 (41.7%)	6 (26.1%)	0.195
Initial weight ^(b)^	59.5 (55–65)	63 (60.5–70.5)	63 (60.5–67)	0.039 *
	^(d)^ vaccinated vs. unvaccinated, *p* = 0.011 *			
	^(d)^ COVID-19 symptoms vs. non-symptoms among vaccinated, *p* = 0.824			
Admission weight ^(b)^	70 (65–78)	74 (69–80.5)	74 (70–78)	0.217
Weight gain in pregnancy ^(b)^	11.5 (8–14)	9 (8–12)	11 (8.5–12)	0.339
BMI ^(b)^	22.46 (21.05–23.77) 2 missing values	24.76 (22.36–25.68)	23.95 (22.46–25.13)	0.012 *
	^(d)^ vaccinated vs. unvaccinated, *p* = 0.004 **			
	^(d)^ COVID-19 symptoms vs. non-symptoms among vaccinated, *p* = 0.482			

^(a)^ mean ± std. dev.; ^(b)^ median (interquartile range with Tukey’s hinges), Kruskal–Wallis statistical test; ^(c)^ observed frequency (percentage); chi-square statistical test (either asymptotic, Fisher’s exact test, or Monte Carlo simulation with 10,000 samples); ^(d)^ Mann–Whitney statistical test; statistical significance: * *p*-value< 0.05; ** *p*-value < 0.01.

**Table 2 medicina-59-00341-t002:** Maternal comorbidities and treatment, stratified by vaccination and symptomatic status.

Comorbidities and Treatments	SARS-CoV-2-Positive Patients			
	Unvaccinated	Vaccinated		
	Symptoms +	Symptoms +	Symptoms −	*p*-Value ^(a)^
Variable ^(a)^	N = 62	N = 12	N = 23	
Presence of at least one comorbidity ^(a)^	38 (61.2%)	7 (58.3%)	17 (73.9%)	0.517
Thyroid disease ^(a)^	4 (6.5%)	−	2 (8.7%)	0.726
Thrombophilia ^(a)^	2 (3.2%)	−	1 (4.3%)	>0.999
Anemia ^(a)^	31 (50.0%)	7 (75%)	15 (65.2%)	0.178
Gestational diabetes ^(a)^	5 (8.1%)	1 (8.3%)	−	0.433
Pregnancy hypertension ^(a)^	9 (14.5%)	−	−	−
Renal disease ^(a)^	−	−	1 (4.3%)	−
Chronic hypertension ^(a)^	−	−	1 (4.3%)	−
Cardiac disease ^(a)^	−	−	1 (4.3%)	−
Iron supplements ^(a)^	17 (27.4%)	−	4 (17.4%)	0.095
Vaginal progesterone ^(a)^	13 (21%)	−	13 (13%)	0.172
Vitamin supplements ^(a)^	18 (29%)	2 (16.7%)	10 (43.5%)	0.260
Aspirin ^(a)^	2 (3.2%)	−	−	−

^(a)^ Observed frequency (percentage); chi-square statistical test (either asymptotic, Fisher’s exact test, or Monte Carlo simulation with 10,000 samples).

**Table 3 medicina-59-00341-t003:** Paraclinical investigations of pregnant patients at admission, stratified by vaccination and symptomatic status. Many investigations had missing values (for each, the actual number of values is specified).

Investigations	SARS-CoV-2-Positive Patients			
	Unvaccinated	Vaccinated		
	Symptoms +	Symptoms +	Symptoms −	*p*-Value ^(a),^
Variable ^(a)^	N = 62	N = 12	N = 23	
Leucocytes ^(a)^	n = 62	n = 11	n = 22	0.691
	9.300 (7.570–12.500)	9.500 (8.550–11.390)	10.650 (9.100–11.700)	
Thrombocytes ^(a)^	n = 62	n = 11	n = 22	0.649
	215,000 (172,000–269,000)	236,000 (189,500–266,000)	213,500.50 (187,000–240,000)	
PCR ^(a)^	n = 50	n = 11	n = 17	0.005 **
	14.15 (8.90–69.80)	44.00 (18.80–73.60)	8.90 (6.80–14.60)	
D-Dimer ^(a)^	n = 38	n = 10	n = 9	0.386
	915.50 (450.00–1361.00)	851.50 (497.00–1125.00)	525.00 (394.00–1013.00)	

^(a)^ Median (interquartile range with Tukey’s hinges), Kruskal–Wallis statistical test; statistical significance: ** *p*-value < 0.01.

**Table 4 medicina-59-00341-t004:** Respiratory management of maternal COVID-19 disease, stratified by vaccination status and symptomatic status.

COVID-19-Related Complications	SARS-CoV-2-Positive Patients			
	Not vaccinated	Vaccinated		
	Symptoms +	Symptoms +	Symptoms −	*p*-Value ^(a)^
Variable ^(a)^	N = 62	N = 12	N = 23	
High flow oxygen therapy ^(a)^	7 (11.3%)	1 (8.3%)	−	0.236
Extracorporeal membrane oxygenation ^(a)^	2 (3.2%)	−	−	0.690
ICU transfer ^(a)^	12 (19.4%)	1 (16.7%)	−	0.070
Endo-tracheal Intubation days ^(a)^				−
2 days	1 (1.6%)	−	−	
8 days	1 (1.6%)	−	−	
15 days	2 (3.2%)	−	−	
25 days	1 (1.6%)	−	−	
28 days	1 (1.6%)	−	−	
29 days	1 (1.6%)	−	−	

^(a)^ Observed frequency (percentage); chi-square statistical test (either asymptotic, Fisher’s exact test, or Monte Carlo simulation with 10,000 samples).

**Table 5 medicina-59-00341-t005:** Obstetrical and neonatal outcome stratified by vaccination and symptomatic status. Except for one case, maternal death and IUFD did not occur in the same patients.

Delivery and Newborn	SARS-CoV-2-Positive Patients			
	Not Vaccinated	Vaccinated		
	Symptoms +	Symptoms +	Symptoms −	*p*-Value ^(a),(b)^
Variable ^(a),(b)^	N = 62	N = 12	N = 23	
Gestational age at delivery ^(a)^	38 (36–40)	38.5 (34–39.5)	39 (38–39.5)	0.747
Prematurity ^(b)^	16 (25.8%)	4 (33.3%)	5(21.7%)	0.886
<28 weeks ^(b)^	2 (3%)	−	−	
28–32 weeks ^(b)^	7 (11.3%)	1 (8.3%)	3 (13%)	
32–37 weeks ^(b)^	7 (11.3%)	3 (25%)	2 (8.69%)	
Term delivery ^(b)^	46(74.1%)	8 (66.6%)	18 (78.2%)	
Gender F ^(b)^	33 (53.2%)	5 (41.7%)	10 (43.5%)	0.615
Maternal death ^(b)^	5 (8.1%)	−	−	−
IUFD ^(b)^	3 (4%)	−	−	−
C-section ^(b)^	40 (64.5%)	10 (83.3%)	15 (65.2%)	0.481
Emergency C−section due to COVID-19 complications	15 (28.8%)	1 (8.3%)	−	0.007 **
Newborn COVID-19 ^(b)^	2 (2.32%)	−	−	−
Birth weight [g] ^(a)^	3150 (2610–3380)	3495 (2590–3670)	3310 (2750–3600)	0.285
5 min Apgar Index ^(a)^	9 (8–9) for 61 newborns 1 missing value	8 (7–9)	8 (8–9)	0.438

^(a)^ Median (interquartile range with Tukey’s hinges), Kruskal–Wallis statistical test; ^(b)^ observed frequency (percentage); chi-square statistical test (either asymptotic, Fisher’s exact test, or Monte Carlo simulation with 10,000 samples); statistical significance: ** *p*-value < 0.01; Abbreviations: Gender F, female; g, grams; IUFD, intrauterine fetal demise.

## Data Availability

Raw data were retrieved from the patients’ records and can be made available only upon formal request and approval of the hospital management and ethics committee.

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
