# Peer review of "Pregnancy Outcomes in SARS-CoV-2-Positive Patients: A 20-Month Retrospective Analysis of Delivery Cases"

_medicina, 2023, doi:10.3390/medicina59020341_

Round 1
Reviewer 1 Report (Previous Reviewer 1)
Dear authors
I read your manuscript and it will help cilincians about the Covid patients management.
Best regards
Author Response
Thank you for taking the time to review.Reviewer 2 Report (Previous Reviewer 3)
Dear Authors
I read the paper: "Pregnancy Outcomes in SARS‑CoV‑2 Positive Patients: a 20-2 Month Retrospective Analysis of Delivery Cases", which falls whithin the aim of Medicina. Honestly, the topic is interesting enough to attract the readers' attention, but the article would benefit from an additional minor revision. I have some additional specific recommendations that I describe under each section:
INTRODUCTION
1)LINE 49: "Three years have passed since the first case of infection with the SARS-CoV-2 coronavirus was reported"
The introduction section should be concise and precise, and this paragraph is redundant. I suggest to delete this line.
2)LINE 50: "The COVID-19 disease has spread globally, increasing morbidity and mortality, and also affecting pregnant women". I suggest to cite novel and influential articles:
- https://doi.org/10.1016/j.ajog.2020.04.030
- https://doi.org/10.1002/ijgo.13726
3)Line 92: Considering the healthcare services and the covid impact on maternity and neonatal care. I suggest to cite novel and related articles:
- https://doi.org/10.1016/S2214-109X(20)30483-6
- https://doi.org/10.1111/jog.15205
4)Line 101: Because the study appeals to an international audience, the study's objective should be well emphasized.
Materials and Methods
5) Exclusion criteria should be better clarified. Did the authors exclude the incomplete data?
RESULTS
6) Line 163: "COVID-19 symptoms in SARS-CoV-2 positive women"
The authors did not report in the Materials and Methods section the COVID-19 symptoms analyzed.
I recommend describing the symptoms included in this retrospective study.
In conclusion, the paper is acceptable with moderate importance and confirms previous work.
I suggest a minor revision.
Author Response
INTRODUCTION
1)LINE 49: "Three years have passed since the first case of infection with the SARS-CoV-2 coronavirus was reported"
The introduction section should be concise and precise, and this paragraph is redundant. I suggest to delete this line.
Answer: Thank you for the suggestion! The line was deleted.
2)LINE 50: "The COVID-19 disease has spread globally, increasing morbidity and mortality, and also affecting pregnant women". I suggest to cite novel and influential articles:
- https://doi.org/10.1016/j.ajog.2020.04.030
- https://doi.org/10.1002/ijgo.13726
Answer:Thank you for the suggestion! The reference was modified
3)Line 92: Considering the healthcare services and the covid impact on maternity and neonatal care. I suggest to cite novel and related articles:
Answer: Thank you for the suggestion! The reference was modified.
4)Line 101: Because the study appeals to an international audience, the study's objective should be well emphasized.
Answer:Thank you for the suggestion! The study’s objectives were emphasized in the text.
Materials and Methods
5) Exclusion criteria should be better clarified. Did the authors exclude the incomplete data?
Answer: Thank you for the suggestion!The study’s criteria were clarified in the text.
Since missing data was present mostly in the part pertaining to blood test investigations, we opted not to exclude it. In the investigations of one transferred patient, missing data (incomplete blood count) was recorded. Additionally, not all patients were sampled for PCR and D-Dimers before birth, which statistically speaking could be regarded as missing data. Absent 5-minute Apgar Score was absent in the records of one transferred patient, and absent ofprepregnanacy BMI was absent in the documentation of two transferred patient.
RESULTS
6) Line 163: "COVID-19 symptoms in SARS-CoV-2 positive women"
The authors did not report in the Materials and Methods section the COVID-19 symptoms analyzed.
Answer: Thank you for the suggestion!Details related to symptoms were added in the text.
Reviewer 3 Report (New Reviewer)
I thank the academic editor for giving me the opportunity to review this manuscript entitled: "Pregnancy Outcomes in SARS-CoV-2 Positive Patients: a 20-Month Retrospective Analysis of Delivery Cases." This is a retrospective, cross-sectional study of 95 patients who were followed in terms of pregnancy and neonatal outcomes. I think the study design is well done and there are some suggestions I can make.
Introduction: this section is well written and clear. I suggest to the authors to improve this section with some words related to gynecological effects of SARS-CoV-2. I suggest these papers:
1) Resta L, Vimercati A, Cazzato G, Mazzia G, Cicinelli E, Colagrande A, Fanelli M, Scarcella SV, Ceci O, Rossi R. SARS-CoV-2 and Placenta: New Insights and Perspectives. Viruses. 2021 Apr 21;13(5):723. doi: 10.3390/v13050723. PMID: 33919284; PMCID: PMC8143362.
2)Carp-Veliscu A, Mehedintu C, Frincu F, Bratila E, Rasu S, Iordache I, Bordea A, Braga M. The Effects of SARS-CoV-2 Infection on Female Fertility: A Review of the Literature. Int J Environ Res Public Health. 2022 Jan 16;19(2):984. doi: 10.3390/ijerph19020984. PMID: 35055804; PMCID: PMC8775865.
3) Dellino M, Lamanna B, Vinciguerra M, Tafuri S, Stefanizzi P, Malvasi A, Di Vagno G, Cormio G, Loizzi V, Cazzato G, Tinelli R, Cicinelli E, Pinto V, Daniele A, Maiorano E, Resta L, De Vito D, Scacco S, Cascardi E. SARS-CoV-2 Vaccines and Adverse Effects in Gynecology and Obstetrics: The First Italian Retrospective Study. Int J Environ Res Public Health. 2022 Oct 13;19(20):13167. doi: 10.3390/ijerph192013167. PMID: 36293746; PMCID: PMC9603573.
Material and methods: it's fine
Results: clear and complete
Discussion: please, I suggest to improve the discussion with concepts related to placental features of the pregnancy women.
Author Response
I thank the academic editor for giving me the opportunity to review this manuscript entitled: "Pregnancy Outcomes in SARS-CoV-2 Positive Patients: a 20-Month Retrospective Analysis of Delivery Cases." This is a retrospective, cross-sectional study of 95 patients who were followed in terms of pregnancy and neonatal outcomes. I think the study design is well done and there are some suggestions I can make.
Introduction: this section is well written and clear. I suggest to the authors to improve this section with some words related to gynecological effects of SARS-CoV-2. I suggest these papers:
1) Resta L, Vimercati A, Cazzato G, Mazzia G, Cicinelli E, Colagrande A, Fanelli M, Scarcella SV, Ceci O, Rossi R. SARS-CoV-2 and Placenta: New Insights and Perspectives. Viruses. 2021 Apr 21;13(5):723. doi: 10.3390/v13050723. PMID: 33919284; PMCID: PMC8143362.
2)Carp-Veliscu A, Mehedintu C, Frincu F, Bratila E, Rasu S, Iordache I, Bordea A, Braga M. The Effects of SARS-CoV-2 Infection on Female Fertility: A Review of the Literature. Int J Environ Res Public Health. 2022 Jan 16;19(2):984. doi: 10.3390/ijerph19020984. PMID: 35055804; PMCID: PMC8775865.
3) Dellino M, Lamanna B, Vinciguerra M, Tafuri S, Stefanizzi P, Malvasi A, Di Vagno G, Cormio G, Loizzi V, Cazzato G, Tinelli R, Cicinelli E, Pinto V, Daniele A, Maiorano E, Resta L, De Vito D, Scacco S, Cascardi E. SARS-CoV-2 Vaccines and Adverse Effects in Gynecology and Obstetrics: The First Italian Retrospective Study. Int J Environ Res Public Health. 2022 Oct 13;19(20):13167. doi: 10.3390/ijerph192013167. PMID: 36293746; PMCID: PMC9603573.
Material and methods: it's fine
Results: clear and complete
Discussion: please, I suggest to improve the discussion with concepts related to placental features of the pregnancy women.
This manuscript is a resubmission of an earlier submission. The following is a list of the peer review reports and author responses from that submission.
Round 1
Reviewer 1 Report
Dear Authors
First of all, it was pleasing to have the opportunity to read about the pregnancy and covid experiences of a different center. I and my team had to treat about 200 Covid 19 positive pregnant women and follow their pregnancies during this bad process. I would like to thank the researchers for their courageous health work for this important group of patients during the Covid 19 process. My questions about the study are listed below.
Comment 1
The available data show that the Covid 19 test is positive in pregnant women who do not show symptoms. This is also included in the review of Pashaei et al., which is a compilation of 238 studies in which pregnancy and prenatal outcomes were evaluated. While symptom positive and negative results of vaccinated patients were evaluated in the study, pregnant women with positive symptom negative test results were not evaluated. These patients were excluded from the study. If left, what is the reason?
Pashaei Z, SeyedAlinaghi S, Qaderi K, Barzegary A, Karimi A, Mirghaderi SP, Mirzapour P, Tantuoyir MM, Dadras O, Ali Z, Voltarelli F, Mehraeen E. Prenatal and neonatal complications of COVID-19: A systematic review. Health Sci Rep. 2022 Feb 15;5(2):e510. doi: 10.1002/hsr2.510.
Comment 2
Were both doses of vaccination completed in pregnant women whose vaccination was completed? How long was the interval between vaccination? How was the situation of the pregnant women who caught the disease before the vaccination process was completed? Have different results been obtained from pregnant women who received full vaccination?
Comment 3
How were the type of anesthesia chosen in pregnant women who underwent C section? Have any patients undergone emergency C-section in ICU patients during the follow-up during the Covid 19 process?

Reviewer 2 Report
The authors present in interesting retrospective analysis of their COVID-19 cases in pregnant women. They could clearly show the very effective protection by vaccination of the pregnant women. That is the most important message - especially because of the the relative low part of vaccination in this cohort.
Because within the vaccinated women two third was asymptomatic - what were the reasons for clinic admission? Please specify.
There was a really high number of maternal death with 8%, but only 2 women with ECMO therapy. Please explain the individual reasons for mortality more detail and which therapy they got before.
Reviewer 3 Report
Dear Authors
I read the paper: "Pregnancy Outcomes in SARS‑CoV‑2 Positive Patients: a 20-2 Month Retrospective Analysis of Delivery Cases", which falls whithin the aim of medicina. The topic is interesting enough to attract the readers' attention, but the article would benefit from a minor revision.
I have some specific recommendations around the study methodology and reporting that I describe under each section:
INTRODUCTION
1) LINE 58: "Pregnancy complications can occur in COVID-19 patients, such as maternal or fetal death, pre-term birth, or caesarian section (C‒section) due to the maternal medical condition"; The authors should improve this section by citing relevant and novel articles about the topic. Please cite these studies in this area, DOI:
- DOI: https://doi.org/10.1002/ijgo.13726
- https://doi.org/10.1016/j.ejogrb.2020.07.034;
2) Line 90: "this worldwide pandemic crisis, our hospital as a whole and staff at all levels faced disruptive challenges in their efforts to continue providing the healthcare services and preserving safe maternity and neonatal care."
The authors should add a paragraph about the lack of maternal care due to the pandemic crisis. To write this section, I suggest to read and cite the following paper, DOI:
- https://doi.org/10.1002/ijgo.13457
- https://doi.org/10.1111/jog.15205
- https://doi.org/10.1186/s12978-021-01070-6
RESULT AND DISCUSSION
3)Researchers have put together the result and discussion section very well.
This paper can be read fluently and engagingly. The title reflects the focus of the analysis. The paper's theoretical framework is very clear, and it fits into the literature against the covid-19.